# On Robust Multiclass Learnability

**Jingyuan Xu**
School of Computer Science
Wuhan University
jingyuanxu777@gmail.com

**Weiwei Liu**[*]
School of Computer Science
Wuhan University
liuweiwei863@gmail.com

## Abstract

This work analyzes the robust learning problem in the multiclass setting. Under the framework of Probably Approximately Correct (PAC) learning, we first show that the graph dimension and the Natarajan dimension, which characterize the standard multiclass learnability, are no longer applicable in robust learning problem. We then generalize these notions to the robust learning setting, denoted as the adversarial graph dimension (AG-dimension) and the adversarial Natarajan dimension (AN-dimension). Upper and lower bounds of the sample complexity of robust multiclass learning are rigorously derived based on the AG-dimension and AN-dimension, respectively. Moreover, we calculate the AG-dimension and AN-dimension of the class of linear multiclass predictors, and show that the graph (Natarajan) dimension is of the same order as the AG(AN)-dimension. Finally, we prove that the AG-dimension and AN-dimension are not equivalent.

## 1 Introduction

Learning models that are robust to adversarial perturbations has attracted significant research attention in recent years[1, 2, 3]. Notably, prior work on this subject has largely studied the theory of robust learnability in the context of binary supervised learning. However, many important problems require classification into a great number of target classes. For example, in image object recognition and language models building[4, 5, 6, 7, 8, 9], the number of classes scales as the number of possible objects or the dictionary size respectively. It is thus of both practical and theoretical interest to generalize the robust binary learning theory to the robust multiclass setting.

In the standard multiclass learning setting, the finiteness of the Natarajan dimension or graph dimension [10] (both of which are generalized from the VC-dimension [11]) is necessary and sufficient to ensure learnability [12]. Recent work has shown that the finiteness of the VC-dimension is neither sufficient nor necessary for (proper) robust binary classification [13]. Inspired by [13], this paper demonstrates that the Natarajan/graph dimension can no longer characterize robust learnability in the multiclass problem. Meanwhile, some questions naturally arise: In multiclass setting, how can robust learning be ensured? And what is necessary for robust learning?

To answer these questions, we generalize the corrupted hypothesis classes [14], which arise from standard hypothesis classes in the binary setting in the presence of an adversary, and use them to reduce the robust learning problem to a non-robust one. By considering the graph dimension and Natarajan dimension of the corrupted hypothesis classes—which are defined as the adversarial graph dimension (AG-dimension) and adversarial Natarajan dimension (AN-dimension) respectively—we derive the upper and lower bounds of sample complexity of the robust multiclass learning problem. We then analyze the AG-dimension and the AN-dimension of linear multiclass predictor class, showing that the AG(AN)-dimension and the graph (Natarajan) dimension are of the same order. Since it is proven that the graph dimension and the Natarajan dimension are equivalent when the

---

[*]Corresponding author

36th Conference on Neural Information Processing Systems (NeurIPS 2022).

target class is finite [12], is it possible to use the AG-dimension and AN-dimension to bound each other? Our work suggests that the equivalent relationship does not hold, which is illustrated by constructing a hypothesis class with finite AN-dimension but not robustly learnable.

**Related work.** [15, 16, 17, 18] focus on the robust binary classification problem. [10] characterize the multiclass learnability in non-robust setting. [19, 12, 20] define a large family of notions of dimensions, all of which generalize the VC-dimension and may be used to estimate the sample complexity of multiclass classification. [21] calculate the graph dimension and the Natarajan dimension of linear multiclass predictor class. [22] upper bounds the adversarial risk of linear multiclass classifiers based on Rademacher complexity, which cannot directly derive bounds of sample complexity of general robust multiclass learning problems. The theory on robust multiclass learnability for general hypothesis class is less explored. This work takes some steps towards solving this problem.

## 2 Preliminaries

This section introduces some basic notations and concepts used throughout the present study.

In the remainder of this article, $\mathbb{R}, \mathbb{N}, \mathbb{R}^+$ and $\mathbb{R}^d$ represent the sets of real numbers, natural numbers, non-negative real numbers and $d$-dimensional vectors over $\mathbb{R}$, respectively. We denote the set $\{1, \ldots, n\}$ (for $n \in \mathbb{N}$) by $[n]$. If $A$ and $B$ are sets, we use $B^A$ to denote the collection of all mappings from $A$ to $B$ and $2^A$ to denote the power set of $A$, that is the collection of all subsets of $A$. Let $C = \{c_1, \ldots, c_m\} \subset \mathcal{X}$ and $C' \subset C$ be a subset of $C$, then $I_{C'} \in [m]$ denotes the index set of $C'$. Let $\mathcal{H}$ be a class of functions defined in $\mathcal{X}$, then $\mathcal{H}|_C$ represents the restriction of $\mathcal{H}$ to $C$, that is $\mathcal{H}_C = \{(h(c_1), \ldots, h(c_m)) : h \in \mathcal{H}\}$. We denote the indicator function by $\mathbb{1}(\text{event})$, that is 1 if an event happens and 0 otherwise. Finally, $\| \cdot \|_p$ represents the $\ell_p$ norm.

The *robust learning* problem is formalized as follows. Let $\mathcal{X} = \mathbb{R}^d$ be the instance space and $\mathcal{Y} = \{1, \ldots, k\}$ be the label space. For an unknown distribution $\mathcal{D}$ over $\mathcal{X} \times \mathcal{Y}^2$ and a hypothesis class $\mathcal{H} \subset \mathcal{Y}^{\mathcal{X}}$, the goal of robust learning is to find a function $f \in \mathcal{H}$, based on $n$ labeled training samples $\mathcal{S} = \{(x_i, y_i)\}_{i=1}^n$ drawn independent and identically distributed (i.i.d.) from $\mathcal{D}$, such that under a small perturbation $\mathcal{U} : \mathcal{X} \to 2^{\mathcal{X}}$, the *adversarial risk*

$$R_{\mathcal{U}}(f; \mathcal{D}) \triangleq \mathbb{E}_{(x,y) \sim \mathcal{D}} \left[ \sup_{x' \in \mathcal{U}(x)} l(f(x'), y) \right]$$

is minimal for the 0-1 loss $l(\hat{y}, y) \triangleq \mathbb{1}(\hat{y} \neq y)$.

The perturbation $\mathcal{U}(x)$ is required to be nonempty, so some choice of $x'$ is always available. One choice for the perturbation is the $p$-norm ball ($p \geq 1$) with a small radius $r$, i.e. $\mathcal{U}(x) = \{z \in \mathcal{X} : \|z - x\|_p \leq r\}$. Selecting $r = 0$ gives the *identity perturbation*: $\mathcal{I}(x) \triangleq \{x\}$. Note that in this case, the problem is reduced to a standard learning problem, and $R_{\mathcal{I}}(f; \mathcal{D})$ is called the *standard risk* of $f$ over $\mathcal{D}$.

The mechanism of finding a function is called a *learning algorithm*. In this paper, we focus on *proper* learning algorithms, that is, learning algorithms will always pick a function from $\mathcal{H}$. Formally, a learning algorithm is a function $\mathcal{A} : \cup_{n=0}^{\infty} (\mathcal{X} \times \mathcal{Y})^n \to \mathcal{H}$. A common choice of robust learning algorithm is to learn through *adversarial empirical risk minimization* or AERM for short:

$$\hat{f} \in \text{AERM}_{\mathcal{U}}(\mathcal{H}; \mathcal{S}) \triangleq \arg\min_{f \in \mathcal{H}} R_{\mathcal{U}}(f; \hat{\mathcal{D}}_n),$$

where $\hat{\mathcal{D}}_n$ is the *empirical distribution* generated by $\mathcal{S}$, i.e. for $(x, y) \sim \hat{\mathcal{D}}_n, (x, y)$ is equal to $(x_i, y_i)$ with probability $1/n$ for each $i \in \{1, \ldots, n\}$.

Learning algorithms like AERM do not always produce a function that achieves the optimal adversarial risk, because a training set consisting of finite samples is not always sufficiently informative to represent $\mathcal{D}$. Thus, we introduce the definition of robust probably approximately correct (PAC) learnability, in the realizable and agnostic setting [14, 13]:

**Definition 1** (Agnostic Robust PAC Learnability). *Under perturbation $\mathcal{U}$, a hypothesis class $\mathcal{H}$ is robustly PAC learnable in the agnostic setting if there is a function $m_{\mathcal{H}, \mathcal{U}}^{ag} : (0, 1)^2 \to \mathbb{N}$ and a*

---

[2]Formally, there is a sigma algebra $\mathcal{F} \subset 2^{\mathcal{X} \times \mathcal{Y}}$ of events and $\mathcal{D}$ is a probability measure on $(\mathcal{X} \times \mathcal{Y}, \mathcal{F})$.

*learning algorithm* $\mathcal{A} : \cup_{n=0}^{\infty}(\mathcal{X} \times \mathcal{Y})^n \to \mathcal{H}$ *with the following property. For every* $\epsilon, \delta \in (0, 1)$ *and every distribution* $\mathcal{D}$ *over* $\mathcal{X} \times \mathcal{Y}$, *let* $\mathcal{S} = \{(x_i, y_i)\}_{i=1}^n$ *be* $n$ *samples i.i.d generated by* $\mathcal{D}$ *s.t.* $n > m_{\mathcal{H},\mathcal{U}}^{ag}(\epsilon, \delta)$. *Then with probability at least* $1 - \delta$,

$$R_{\mathcal{U}}(\mathcal{A}(\mathcal{S}); \mathcal{D}) \leq \min_{f \in \mathcal{H}} R_{\mathcal{U}}(f; \mathcal{D}) + \epsilon.$$

*The function* $m_{\mathcal{H},\mathcal{U}}^{ag}$ *is defined as the sample complexity of learning* $\mathcal{H}$ *under perturbation* $\mathcal{U}$ *in the agnostic setting.*

**Definition 2** (Realizable Robust PAC Learnability)**.** *Under perturbation* $\mathcal{U}$, *a hypothesis class* $\mathcal{H}$ *is robustly PAC learnable in the realizable setting if there is a function* $m_{\mathcal{H},\mathcal{U}}^{re} : (0, 1)^2 \to \mathbb{N}$ *and a learning algorithm* $\mathcal{A} : \cup_{n=0}^{\infty}(\mathcal{X} \times \mathcal{Y})^n \to \mathcal{H}$ *with the following property. For every* $\epsilon, \delta \in (0, 1)$ *and every distribution* $\mathcal{D}$ *over* $\mathcal{X} \times \mathcal{Y}$, *let* $\mathcal{S} = \{(x_i, y_i)\}_{i=1}^n$ *be* $n$ *samples i.i.d generated by* $\mathcal{D}$ *s.t.* $n > m_{\mathcal{H},\mathcal{U}}^{re}(\epsilon, \delta)$. *Then with probability at least* $1 - \delta$,

$$R_{\mathcal{U}}(\mathcal{A}(\mathcal{S}); \mathcal{D}) \leq \epsilon.$$

*The function* $m_{\mathcal{H},\mathcal{U}}^{re}$ *is defined as the sample complexity of learning* $\mathcal{H}$ *under perturbation* $\mathcal{U}$ *in the realizable setting.*

These definitions agree with the standard PAC learnability when $\mathcal{U} = \mathcal{I}$, which has been widely studied in both *binary* ($k = 2$) and *multiclass* ($k > 2$) setting. We recall the known result regarding the sample complexity of binary learning. Recall the definition of the Vapnik-Chervonenkis dimension (VC-dimension) [11]:

**Definition 3** (VC-dimension)**.** *Let* $\mathcal{H} \subset \{0, 1\}^{\mathcal{X}}$ *be a hypothesis class and let* $S \subset \mathcal{X}$. *We say that* $S$ *is shattered by* $\mathcal{H}$ *if* $\mathcal{H}|_S = \{0, 1\}^S$. *The VC-dimension of* $\mathcal{H}$, *denoted by* $\mathrm{VC}(\mathcal{H})$, *is the maximal cardinality of a set that is shattered by* $\mathcal{H}$.

The following theorem, based on the VC-dimension, characterizes the sample complexity of learning *binary* hypothesis classes.

**Theorem 1** ([11] and [23])**.** *There are absolute constants* $C_1, C_2 > 0$ *such that for every* $\mathcal{H} \subset \{0, 1\}^{\mathcal{X}}$,

$$C_1 \left( \frac{\mathrm{VC}(\mathcal{H}) + \ln \frac{1}{\delta}}{\epsilon} \right) \leq m_{\mathcal{H},\mathcal{I}}^{re}(\epsilon, \delta) \leq C_2 \left( \frac{\mathrm{VC}(\mathcal{H}) \ln \frac{1}{\epsilon} + \ln \frac{1}{\delta}}{\epsilon} \right),$$

*and*

$$C_1 \left( \frac{\mathrm{VC}(\mathcal{H}) + \ln \frac{1}{\delta}}{\epsilon^2} \right) \leq m_{\mathcal{H},\mathcal{I}}^{ag}(\epsilon, \delta) \leq C_2 \left( \frac{\mathrm{VC}(\mathcal{H}) + \ln \frac{1}{\delta}}{\epsilon^2} \right).$$

It is natural to seek a generalization of the VC-dimension to hypothesis classes of non-binary functions. We recall two generalizations, both introduced by [10]:

**Definition 4** (Graph dimension)**.** *Let* $\mathcal{H} \subset \mathcal{Y}^{\mathcal{X}}$ *be a hypothesis class and let* $S \subset \mathcal{X}$. *We say that* $\mathcal{H}$ *G-shatters* $S$ *if there exists an* $f : S \to \mathcal{Y}$ *such that for every* $T \subset S$, *there is a* $g \in \mathcal{H}$ *such that*

$$\forall x \in T, g(x) = f(x), \quad and \quad \forall x \in S \backslash T, g(x) \neq f(x).$$

*The graph dimension of* $\mathcal{H}$, *denoted by* $d_G(\mathcal{H})$, *is the maximal cardinality of a set that is G-shattered by* $\mathcal{H}$.

**Definition 5** (Natarajan dimension)**.** *Let* $\mathcal{H} \subset \mathcal{Y}^{\mathcal{X}}$ *be a hypothesis class and let* $S \subset \mathcal{X}$. *We say that* $\mathcal{H}$ *N-shatters* $S$ *if there exists* $f_1, f_2 : S \to \mathcal{Y}$ *such that* $\forall x \in S, f_1(x) \neq f_2(x)$, *and for every* $T \subset S$, *there is a* $g \in \mathcal{H}$ *such that*

$$\forall x \in T, g(x) = f_1(x), \quad and \quad \forall x \in S \backslash T, g(x) = f_2(x).$$

*The Natarajan dimension of* $\mathcal{H}$, *denoted by* $d_N(\mathcal{H})$, *is the maximal cardinality of a set that is N-shattered by* $\mathcal{H}$.

Both the graph dimension and the Natarajan dimension coincide with the VC-dimension for $k = 2$. Based on these notions, the following theorem provides the upper and lower bounds of sample complexity of standard multiclass learning:

**Theorem 2** ([10] and [12]). *There are absolute constants $C_1, C_2 > 0$ such that for every $\mathcal{H} \subset \mathcal{Y}^{\mathcal{X}}$,*

$$C_1\left(\frac{d_N(\mathcal{H}) + \ln\frac{1}{\delta}}{\epsilon}\right) \leq m^{re}_{\mathcal{H},\mathcal{I}}(\epsilon, \delta) \leq C_2\left(\frac{d_G(\mathcal{H})\ln\frac{1}{\epsilon} + \ln\frac{1}{\delta}}{\epsilon}\right),$$

*and*

$$C_1\left(\frac{d_N(\mathcal{H}) + \ln\frac{1}{\delta}}{\epsilon^2}\right) \leq m^{ag}_{\mathcal{H},\mathcal{I}}(\epsilon, \delta) \leq C_2\left(\frac{d_G(\mathcal{H}) + \ln\frac{1}{\delta}}{\epsilon^2}\right).$$

## 3 AG-dimension and upper bounds for robust multiclass learning

We start by showing that finite graph dimension is not sufficient for robust multiclass learnability. We then introduce the notion of corrupted hypotheses, presented by [14]. By computing the graph dimension of these hypotheses, which is defined as adversarial graph dimension, we can upper bound the sample complexity of robust multiclass learning problem in both realizable and agnostic settings.

### 3.1 Finite graph dimension is not sufficient for robust multiclass learning

Theorem 2 shows that the finiteness of the graph dimension is a sufficient condition for multiclass learnability in non-robust setting. While in adversarial setting, for hypothesis classes with finite graph dimension, one cannot ensure the robust PAC learnability of $\mathcal{H}$, even if $d_G(\mathcal{H}) = 1$.

**Theorem 3.** *There exists a hypothesis class $\mathcal{H}$ with $d_G(\mathcal{H}) = 1$ and an adversary $\mathcal{U}$ such that $\mathcal{H}$ is not robustly PAC learnable with respect to $\mathcal{U}$.*

To prove this theorem, We first present a lemma, which is generalized from Lemma 3 in [13].

**Lemma 1.** *Let $m \in \mathbb{N}$. Then, there exists a hypothesis class $\mathcal{H} \subset \mathcal{Y}^{\mathcal{X}}$ such that for any learning rule $\mathcal{A} : \cup_{n=0}^{\infty}(\mathcal{X} \times \mathcal{Y})^n \to \mathcal{H}$, there exists a distribution $\mathcal{D}$ over $\mathcal{X} \times \mathcal{Y}$ such that:*

*1. There exists a function $f^* \in \mathcal{H}$ with $R_{\mathcal{U}}(f^*; \mathcal{D}) = 0$.*

*2. With probability of at least $1/7$ over the choice of $\mathcal{S} \sim \mathcal{D}^m$ we have that $R_{\mathcal{U}}(\mathcal{A}(\mathcal{S}); \mathcal{D}) \geq 1/8$.*

The proof of this lemma is presented in Appendix. We now proceed with the proof of Theorem 3.

*Proof of Theorem 3.* The argument follows closely a proof of an analogous result by [13] for binary robust learning, but generalizes the constructions and analyzes to match the definition of graph dimension in the multiclass setting. For each $m \in \mathbb{N}$, let $X_m = \{x_1^{(m)}, \ldots, x_{3m}^{(m)}\}$ be a set that contains $3m$ distinct points in $\mathcal{X}$ subject to $\forall x_i, x_j \in \cup_{m=1}^{\infty} X_m$, if $x_i \neq x_j$, then $\mathcal{U}(x_i) \cap \mathcal{U}(x_j) = \emptyset$. And we construct $\mathcal{H}_m$ on $X_m$ as follows. For each $b \in \{0,1\}^{3m}$, we construct a set $\mathcal{Z}_b$ : Initialize $\mathcal{Z}_b = \emptyset$, for each $i \in [3m]$, if $b_i = 1$ then pick a point $z \in \mathcal{U}(x_i^{(m)})$ such that $z \notin \mathcal{Z}_{b'}$ for each $b' \neq b$, and add it to $\mathcal{Z}_b$. Let $h_b : \mathcal{X} \to \mathcal{Y}$ be a hypothesis such that $h_b(x) = 1$ if and only if $x \notin \mathcal{Z}_b$ and $x \notin X_{m'}$ for $m' \neq m$. Furthermore, for fixed $m \in \mathbb{N}$ and $x \in X_{m'}$ s.t. $m' \neq m$, we need all $h_b$ map $x$ into a same label $y_x(\neq 1)$, where $y_x$ is consistent for all $m$. That is,

$$h_b(x) = \begin{cases} \neg 1, & x \in Z_b, \\ y_x(\neq 1), & \exists m' \neq m, x \in X_{m'}, \\ 1, & \text{otherwise,} \end{cases}$$

where $\neg 1$ is some label that is not equal to 1. Then define

$$\mathcal{H}_m = \{h_b : b \in \{0,1\}^{3m}, \sum_{i=1}^{3m} b_i = m\}.$$

Let $\mathcal{H} = \cup_{m=1}^{\infty} \mathcal{H}_m$. We claim that $d_G(\mathcal{H}) \leq 1$. That is, if pick two points $x_1, x_2 \in \mathcal{X}$ such that $x_1 \neq x_2$, $\mathcal{H}$ cannot G-shatters $\{x_1, x_2\}$. since for all $h \in \mathcal{H}$ and $x \in \mathcal{X}$, there are at most two cases of $h(x)$, one of which is 1. Hence the only $f : \{x_1, x_2\} \to \mathcal{Y}$ in Definition 4 that needs to be considered is $f(x_1) = f(x_2) = 1$. There are six cases to consider:
(1) There exists $m \in \mathbb{N}$ such that $z_1, z_2 \in X_m$. Then $\forall h_b \in \mathcal{H}_m, x_1, x_2 \notin Z_b$ by the construction of $Z_b$. Hence $h_b(x_1) = h_b(x_2) = 1$. And $\forall h \in \mathcal{H}_{m'}$ for $m' \neq m$, we have $h(x_1) = y_{x_1} \neq 1, h(x_2) = y_{x_2} \neq 1$. So we only obtain labelings $(1, 1)$ and $(\neg 1, \neg 1)$.

(2) There exists $m \in \mathbb{N}$ such that $x_1, x_2 \in \mathcal{U}(X_m) \backslash X_m$. There are two sub-cases to consider. (i) $\exists b$ s.t. $h_b \in \mathcal{H}_m$ and $x_1, x_2 \in Z_b$, then we have $h_b(x_1) \neq 1$ and $h_b(x_2) \neq 1$. For the remaining $h \in \mathcal{H}$ we have $h(x_1) = h(x_2) = 1$. So we only obtain labelings $(1, 1)$ and $(\neg 1, \neg 1)$ in this sub-case. (ii) For the remaining sub-cases, labeling $(\neg 1, \neg 1)$ cannot be obtained since $x_1, x_2 \notin X_m$ for all $m$ and there is no $b$ such that $x_1, x_2$ are in $Z_b$ in the same time.
(3) $x_1 \in X_m$ and $x_2 \in X_{m'}$ for $m \neq m'$. In this case, we have $h(x_1) = 1, h(x_2) \neq 1$ for $h \in \mathcal{H}_m$, and $h(x_1) \neq 1, h(x_2) = 1$ for $h \in \mathcal{H}_{m'}$. For those predictors that are in neither $\mathcal{H}_m$ nor $\mathcal{H}_{m'}$, we have $h(x_1) \neq 1, h(x_2) \neq 1$. Hence we cannot label both points $x_1$ and $x_2$ with $(1, 1)$.
(4) $x_1 \in X_m$ and $x_2 \in \mathcal{U}(X_{m'}) \backslash X_{m'}$ for $m \neq m'$. Observe that $h(x_1) = 1$ if and only if $h \in \mathcal{H}_m$. But for all $h \in \mathcal{H}_m, h$ always labels $x_2$ with 1. Hence labeling $(1, \neg 1)$ cannot be obtained.
(5) $x_1 \in \mathcal{U}(X_m) \backslash X_m$ and $x_2 \in \mathcal{U}(X_{m'}) \backslash X_{m'}$ for $m \neq m'$. Similar to the discussion above, for all $h \in \mathcal{H}$, either $h(x_1) = 1$ or $h(x_2) = 1$. So we cannot obtain labeling $(\neg 1, \neg 1)$.
(6) The remaining cases, i.e. $x_1, x_2 \notin \mathcal{U}(X_m)$ for all $m \in \mathbb{N}$. In this case we can only obtain labeling $(1, 1)$ since for all $h \in \mathcal{H}$, we have $h(x_1) = h(x_2) = 1$.

The arguments above show that $d_G(\mathcal{H}) \leq 1$. It remains to show $\mathcal{H}$ is not robustly PAC learnable. By Lemma 1, it follows that for any learning rule $\mathcal{A} : \cup_{n=0}^{\infty} (\mathcal{X} \times \mathcal{Y})^n \to \mathcal{H}$ and for any $m \in \mathbb{N}$, there exists a distribution $\mathcal{D}$ over $X_m \times \mathcal{Y}$ where there exists a predictor $h^* \in \mathcal{H}$ with $R_{\mathcal{U}}(h^*; \mathcal{D}) = 0$, but with probability at least $1/7$ over $\mathcal{S} \sim \mathcal{D}^m$, $R_{\mathcal{U}}(\mathcal{A}(\mathcal{S}); \mathcal{D}) > 1/8$ if $\mathcal{A}(\mathcal{S}) \in \mathcal{H}_m$. Classifiers in $\mathcal{H}_{m'}$ where $m' \neq m$ are also non-robust since they make mistakes on points in $X_m$. This concludes that $\mathcal{H}$ is not robustly PAC learnable, which completes the proof. $\qquad\square$

### 3.2 Adversarial graph dimension

Consider a given hypothesis $f : \mathcal{X} \to \mathcal{Y}$. A labeled adversarial sample $(\tilde{x}, y)$ is classified correctly if $\tilde{x} \in f^{-1}(y)$. A labeled example $(x, y)$ is classified correctly if $\mathcal{U}(x) \subset f^{-1}(y)$. Following [14], let $\tilde{\mathcal{Y}} = \mathcal{Y} \cup \{\bot\}$, where $\bot$ is the special "always wrong" output, i.e. $l(y, \bot) = l(\bot, y) = 1$ for every $y \in \tilde{\mathcal{Y}}$. We define the mapping $\kappa_{\mathcal{U}} : \mathcal{Y}^{\mathcal{X}} \to \tilde{\mathcal{Y}}^{\mathcal{X}}$:

$$\kappa_{\mathcal{U}}(f)(x) = \begin{cases} y, & \mathcal{U}(x) \subset f^{-1}(y), \\ \bot, & \text{otherwise.} \end{cases} \tag{1}$$

The corrupted set of hypotheses induced by perturbation $\mathcal{U}$ is then defined by $\tilde{\mathcal{H}} = \{\kappa_{\mathcal{U}}(f) : f \in \mathcal{H}\}$.

**Lemma 2** (Lemma 2 in [14]). *For any perturbation $\mathcal{U}$ and distribution $\mathcal{D}, R_{\mathcal{U}}(f; \mathcal{D}) = R_{\mathcal{I}}(\kappa_{\mathcal{U}}(f); \mathcal{D})$.*

This lemma bridges the equivalence between robustly learning $\mathcal{H}$ and learning $\tilde{\mathcal{H}}$ without adversary, which enables us to use standard techniques to bound the sample complexity. Intuitively, we can derive a sufficient condition by considering the graph dimension of $\tilde{\mathcal{H}}$, which can be equivalently formalized as follows:

**Definition 6** (Adversarial Graph Dimension). *Let $\mathcal{H} \subset \mathcal{Y}^{\mathcal{X}}$ be a hypothesis class and let $S \subset \mathcal{X}$. We say that $\mathcal{H}$ adversarially G-shatters $S$ if there exists an $f : \mathcal{X} \to \mathcal{Y}$ such that for every $T \subset S$, there is a $g \in \mathcal{H}$ such that*

$$\forall x \in T, \forall x' \in \mathcal{U}(x), g(x') = f(x)$$

*and*

$$\forall x \in S \backslash T, \exists x' \in \mathcal{U}(x), g(x') \neq f(x).$$

*The adversarial graph dimension (AG-dimension) of $\mathcal{H}$, denoted by $d_G^{\mathcal{U}}(\mathcal{H})$, is the maximal cardinality of a set that is adversarially G-shattered by $\mathcal{H}$.*

**Proposition 1.** *Let $\mathcal{H} \subset \mathcal{Y}^{\mathcal{X}}$ be a hypothesis class and let $\tilde{\mathcal{H}}$ be the corrupted set of hypotheses induced by perturbation $\mathcal{U}$. Then we have*

$$d_G(\tilde{\mathcal{H}}) = d_G^{\mathcal{U}}(\mathcal{H}).$$

The proof is presented in Appendix.

Intuitively, by combining the above proposition with Lemma 2 and Theorem 2, we can upper bound the sample complexity of robust multiclass learning. Formally:

**Theorem 4.** *There are absolute constants $C > 0$ such that for every $\mathcal{H} \subset \mathcal{Y}^{\mathcal{X}}$,*

$$m_{\mathcal{H},\mathcal{U}}^{re}(\epsilon, \delta) \leq C\left(\frac{d_G^{\mathcal{U}}(\mathcal{H})\ln(\frac{1}{\epsilon}) + \ln(\frac{1}{\delta})}{\epsilon}\right), \quad and \quad m_{\mathcal{H},\mathcal{U}}^{ag}(\epsilon, \delta) \leq C\left(\frac{d_G^{\mathcal{U}}(\mathcal{H}) + \ln(\frac{1}{\delta})}{\epsilon^2}\right).$$

*Proof.* Let $\mathcal{H} \subset \mathcal{Y}^{\mathcal{X}}$ be a hypothesis class with $d_G^{\mathcal{U}}(\mathcal{H}) = d$. For every $f \in \mathcal{H}$, define $\bar{f} : \mathcal{X} \times \mathcal{Y} \to \{0, 1\}$ by setting $\bar{f}(x, y) = 1$ if and only if $f(x') = y$ for all $x' \in \mathcal{U}(x)$. Note that according to our construction, we have

$$\mathbb{P}_{(x,y)\sim\mathcal{D}}[\bar{f}(x, y) \neq 1] = \mathbb{E}_{(x,y)\sim\mathcal{D}}\left[\sup_{x'\in\mathcal{U}(x)} \mathbb{1}(f(x') \neq y)\right]. \tag{2}$$

In the realizable case, let $\bar{\mathcal{H}} = \{\bar{f} : f \in \mathcal{H}\}$. From the definition, we can see that $\mathrm{VC}(\bar{\mathcal{H}}) = d_G^{\mathcal{U}}(\mathcal{H})$. By Theorem 1, there exists $C > 0$ such that when we draw $m > C(\frac{d}{\epsilon}\ln(\frac{1}{\epsilon}) + \frac{1}{\epsilon}\ln(\frac{1}{\delta}))$ samples i.i.d. from some distribution $\mathcal{D}$ on $\mathcal{X} \times \mathcal{Y}$, then with probability at least $1 - \delta$ we have $\mathbb{P}_{(x,y)\sim\mathcal{D}}[\bar{f}(x, y) \neq 1] \leq \epsilon$. Plugging 2 into this inequality, we complete the proof for the realizable case and it is similar for the agnostic case. □

# 4 AN-dimension and lower bounds for robust multiclass learning

In this section, we discuss the necessary conditions for robust multiclass learnability.

## 4.1 Finite Natarajan dimension is not necessary for robust multiclass learning

We first show that the finiteness of ordinary Natarajan dimension is also not necessary for robust multiclass learnability.

**Theorem 5.** *There exist $\mathcal{H}, \mathcal{U}$ such that $d_N(\mathcal{H}) = \infty$ but $\mathcal{H}$ is robustly PAC learnable under $\mathcal{U}$, i.e. $m_{\mathcal{H},\mathcal{U}}^{ag}(\epsilon, \delta) < \infty$.*

*Proof.* The proof follows the construction from [13], which shows that finite VC-dimension is not necessary for robust binary learnability. Let $\mathcal{H} = \mathcal{Y}^{\mathcal{X}}$ be the family of all mappings from $\mathcal{X}$ to $\mathcal{Y}$ and $\mathcal{U}(x) = \mathcal{X}$ be an all-powerful adversary. It is easy to see that $\forall n \in \mathbb{N}$, there exist $n$ distinct points in $\mathbb{R}^d$ that are N-shattered by $\mathcal{H}$, hence $d_N(\mathcal{H}) = \infty$. Given a distribution $\mathcal{D}$ on $\mathcal{X} \times \mathcal{Y}$, let $\mathcal{S} = \{(x_i, y_i)\}_{i=1}^m$ be $m$ samples drawn i.i.d from $\mathcal{D}$ and $\mathcal{A}$ be a learning rule such that $\mathcal{A}(\mathcal{S})$ returns a constant-label predictor $h$ :

$$h(x) = k_m := \underset{k\in\mathcal{Y}}{\arg\max}\left|\{1 \leq i \leq m : y_i = k\}\right|, \qquad \forall x \in \mathcal{X}.$$

According to the weak law of large numbers, $\hat{\mathcal{D}}_m \to \mathcal{D}$ as $m \to \infty$, where $\hat{\mathcal{D}}_m$ is the empirical distribution generated by $S$, we have $\mathbb{P}\left[k_m = \arg\max_{k\in\mathcal{Y}} \mathbb{P}_{(x,y)\sim\mathcal{D}}[y = k]\right] \to 1$ as $m \to \infty$. Hence for all $\delta \in (0, 1)$ there exists a $m_0 \in \mathbb{N}$ such that when $m > m_0$, $R_{\mathcal{U}}(h; \mathcal{D})$ reaches the optimal adversarial risk in $\mathcal{H}$, i.e. $R_{\mathcal{U}}(h; \mathcal{D}) = \arg\min_{f\in\mathcal{H}} R_{\mathcal{U}}(f; \mathcal{D})$ with probability at least $1 - \delta$. □

## 4.2 Adversarial Natarajan dimension

Following the idea of generalizing the graph dimension to the AG-dimension, for a given hypothesis class $\mathcal{H}$, one can consider the Natarajan dimension of its corrupted hypothesis class to lower bound the sample complexity of robust learnability. However, there are several natural but different ways in which the notion of N-shattering can be generalized: that is, we can either let $f_2$ that witnesses $\tilde{\mathcal{H}}$ N-shattering $S$ take the value " $\perp$ " or not. Both notions derive a necessary condition for robust learnability. In this work, we choose the stronger notion, i.e. we need the entire perturbation sets to be N-shattered by $\mathcal{H}$.

**Definition 7** (Adversarial Natarajan Dimension). *Let $\mathcal{H} \subset \mathcal{Y}^{\mathcal{X}}$ be a hypothesis class and let $S \subset \mathcal{X}$. We say that $\mathcal{H}$ adversarially N-shatters $S$ if there exist $f_1, f_2 : \mathcal{X} \to \mathcal{Y}$ such that $\forall y \in S, f_1(y) \neq f_2(y)$, and for every $T \subset S$, there is a $g \in \mathcal{H}$ such that*

$$\forall x \in T, \forall x' \in \mathcal{U}(x), g(x') = f_1(x)$$

*and*

$$\forall x \in S \backslash T, \forall x' \in \mathcal{U}(x), g(x') = f_2(x).$$

*The adversarial Natarajan dimension (AN-dimension) of $\mathcal{H}$, denoted by $d_N^{\mathcal{U}}(\mathcal{H})$, is the maximal cardinality of a set that is adversarially N-shattered by $\mathcal{H}$.*

Based on this definition, we present that:

**Theorem 6.** *There are absolute constants $C > 0$ such that for every $\mathcal{H} \subset \mathcal{Y}^{\mathcal{X}}$,*

$$m_{\mathcal{H},\mathcal{U}}^{re}(\epsilon, \delta) \geq C \left( \frac{d_N^{\mathcal{U}}(\mathcal{H}) + \ln(\frac{1}{\delta})}{\epsilon} \right), \quad and \quad m_{\mathcal{H},\mathcal{U}}^{ag}(\epsilon, \delta) \geq C \left( \frac{d_N^{\mathcal{U}}(\mathcal{H}) + \ln(\frac{1}{\delta})}{\epsilon^2} \right).$$

*Proof.* Let $\mathcal{H} \subset \mathcal{Y}^{\mathcal{X}}$ be a hypothesis class such that $d_N^{\mathcal{U}}(\mathcal{H}) = d$, and let $\mathcal{H}_d = \{0,1\}^{[d]}$ be the family of all mappings from $[d]$ to $\{0,1\}$. We want to show that for every learning algorithm $\mathcal{A}$ for $\mathcal{H}$ and every $\epsilon, \delta \in (0,1)$, there exists a learning algorithm $\bar{\mathcal{A}}$ satisfying: if there exists a function $m_{\mathcal{A},\mathcal{U}}^{re} : (0,1)^2 \to \mathbb{N}$ such that when $\mathcal{A}$ witnesses $m > m_{\mathcal{A},\mathcal{U}}^{re}(\epsilon, \delta)$ samples (denoted as $S$) drawn i.i.d from some distribution $\mathcal{D}$, $\mathbb{P}[R_{\mathcal{U}}(\mathcal{A}(S); \mathcal{D}) \leq \epsilon] > 1 - \delta$. Then when $\bar{\mathcal{A}}$ witnesses $m$ samples (denoted as $S'$) drawn i.i.d from some distribution $\mathcal{D}'$ on $[d] \times \{0,1\}$, $\mathbb{P}[R_{\mathcal{I}}(\bar{\mathcal{A}}(S'); \mathcal{D}') \leq \epsilon] > 1 - \delta$ holds. By this claim, we have $m_{\bar{\mathcal{A}},\mathcal{I}}^{re} \leq m_{\mathcal{A},\mathcal{U}}^{re}$, thus $m_{\mathcal{H}_d,\mathcal{I}}^{re} \leq m_{\mathcal{H},\mathcal{U}}^{re}$. Note that VC $(\mathcal{H}_d) = d$, so by Theorem 1 we have $m_{\mathcal{H},\mathcal{U}}^{re}(\epsilon, \delta) \geq C(\frac{d + \ln(1/\delta)}{\epsilon})$ for some constant $C > 0$ (independent of $\mathcal{H}$), and similarly for the agnostic case.

We now prove the claim. Let $\{s_1, \ldots, s_d\} \subset \mathcal{X}$ be a set and let $f_0, f_1$ be functions that witness the adversarial N-shattering of $\mathcal{H}$. Given a sample set $S' \triangleq \{(x_i, y_i)\}_{i=1}^m \subset [d] \times \{0,1\}$, let $g = \mathcal{A}(\{s_{x_i}, f_{y_i}(s_{x_i})\}_{i=1}^m)$. Consider the learning algorithm $\bar{\mathcal{A}} : \cup_{n=0}^{\infty}([d] \times \{0,1\})^n \to \mathcal{H}_d$ :

$$\bar{\mathcal{A}}(S')(j) = \begin{cases} 1, & g(s_j') = f_1(s_j), \forall s_j' \in \mathcal{U}(s_j), \\ 0, & \text{otherwise.} \end{cases}$$

Denote $\bar{\mathcal{A}}(S')$ by $f$. By the definition of adversarial N-shattering, it can be derived that $f(j) = 0$ if and only if $g(s_j') = f_2(s_j), \forall s_j' \in \mathcal{U}(s_j)$. By the construction we have

$$\mathbb{E}_{(j,y) \sim \mathcal{D}'}[\mathbb{1}(f(j) \neq y)] = \mathbb{E}_{(j,y) \sim \mathcal{D}'}[\sup_{s_j' \in \mathcal{U}(s_j)} \mathbb{1}(g(s_j') \neq f_y(s_j))] = \mathbb{E}_{(x,y) \sim \hat{\mathcal{D}}_m'} R_{\mathcal{U}}(g; \hat{\mathcal{D}}_m'),$$

where $\hat{\mathcal{D}}_m'$ is defined by $\mathbb{P}_{\hat{\mathcal{D}}_m'}(X = s_j, Y = f_y(s_j)) = \mathbb{P}_{\mathcal{D}'}(J = j, B = y)$. $\qquad \square$

## 5 The AG/AN-dimension of linear multiclass predictors

In this section, we calculate the AG-dimension and AN-dimension for linear multiclass predictors under bounded $l_p$ perturbation, motivated by the binary results of [14] and the non-adversarial results of [21]. Recall the definition of linear multiclass predictors. Let $\Psi : \mathcal{X} \times \mathcal{Y} \to \mathbb{R}^{dk}$ be a class-sensitive feature mapping [24]:

$$\Psi(x, y) = [\underbrace{0, \ldots, 0}_{\in \mathbb{R}^{(y-1)d}}, \underbrace{x^{(1)}, \ldots, x^{(d)}}_{\in \mathbb{R}^d}, \underbrace{0, \ldots, 0}_{\in \mathbb{R}^{(k-y)d}}],$$

where $x^{(i)}$ represents the $i^{\text{th}}$ coordinate of $x$. The class of linear multiclass predictors is defined as

$$\mathcal{H}_\Psi = \{x \mapsto \arg\max_{i \in \mathcal{Y}} \langle w, \Psi(x, i) \rangle : w \in \mathbb{R}^{dk}\}. \tag{3}$$

We cite that the graph dimension and Natarajan dimension of $\mathcal{H}_\Psi$ is given by $(d-1)(k-1) \leq d_N(\mathcal{H}_\Psi) \leq d_G(\mathcal{H}_\Psi) \leq O(dk \ln(dk))$ [21]. We will demonstrate that the AG-dimension and AN-dimension also satisfy this bound in the remainder of this section. To this end, we start by calculating the AN-dimension of $\mathcal{H}_\Psi$. Specifically, one can show that the AN-dimension of $\mathcal{H}_\Psi$ equals to the standard Natarajan dimension of $\mathcal{H}_\Psi$.

**Theorem 7.** *Let $\mathcal{U}(x) = \{z \in \mathcal{X} : \|z - x\|_p \leq r\}$ for some $r \in \mathbb{R}^+$ and $p \in \mathbb{R}^+ \cup \{\infty\}$, and let $\mathcal{H}_\Psi$ be as defined in 3. Then, the AN-dimension of $\mathcal{H}_\Psi$ satisfies $d_N^{\mathcal{U}}(\mathcal{H}_\Psi) = d_N(\mathcal{H}_\Psi)$.*

*Proof.* Since adversarial N-shattering implies N-shattering, we have $d_N^{\mathcal{U}}(\mathcal{H}_\Psi) \leq d_N(\mathcal{H}_\Psi)$. To prove the inverse inequality, it should be noted that the classifiers in $\mathcal{H}_\Psi$ are linear, hence if $\mathcal{H}_\Psi$ N-shatters $S$, it also N-shatters the set with some $x \in S$ replaced by $\forall x' \in l_x$, where $l_x$ is the half-line which terminates at 0 and goes through $x$. For those $x \in S$ that cannot be adversarially N-shattered by $\mathcal{H}_\Psi$, we can always find some $x' \in l_x$ such that $\mathcal{U}(x')$ does not cross the decision boundary of $g_T$ that witnesses the N-shattering for all $T \in S$. $\square$

Note that the Natarajan dimension of $\mathcal{H}_\Psi$ satisfies $(d-1)(k-1) \leq d_N(\mathcal{H}_\Psi) \leq dk$ (Corollary 29.8 in [25]). Combine this result with Theorem 7, we have:

**Corollary 1.** *Under the conditions of Theorem 7,* $(d-1)(k-1) \leq d_N^{\mathcal{U}}(\mathcal{H}_\Psi) \leq dk$.

Next, we present the AG-dimension of $\mathcal{H}_\Psi$:

**Theorem 8.** *Let* $\mathcal{U}(x) = \{z \in \mathcal{X} : \|z - x\|_p \leq r\}$ *for some* $r \in \mathbb{R}^+$ *and* $p \in \mathbb{R}^+ \cup \{\infty\}$, *and let* $\mathcal{H}_\Psi$ *be as defined in 3. Then, the AG-dimension of* $\mathcal{H}_\Psi$ *satisfies*

$$(d-1)(k-1) \leq d_G^{\mathcal{U}}(\mathcal{H}_\Psi) \leq O(dk \ln(dk)).$$

To prove this result, we first present a lemma, which can be viewed as the generalization of Sauer's Lemma [26] in the adversarial setting. We first define the restriction of a corrupted class. Let $\tilde{\mathcal{H}}$ be the corrupted hypothesis class induced by $\mathcal{U}$. Let $S = \{x_1, \dots, x_m\} \subset \mathcal{X}$. Fix a $f \in \mathcal{Y}^S$, for each $h \in \mathcal{H}$, define $h_f : S \to \{0, 1\}$ such that $h_f(x_i) = 1$ if and only if $\kappa_{\mathcal{U}}(h)(x_i) = f(x_i), i \in [m]$. Define $\mathcal{H}_f = \{h_f : h \in \mathcal{H}\}$. The restriction of $\tilde{\mathcal{H}}$ to $S$ is defined by:

$$\tilde{\mathcal{H}}|_S = \mathcal{H}_{f*}|_S, \quad \text{where } f^* = \arg\max_{f \in \mathcal{Y}^S} \left| \mathcal{H}_f|_S \right|.$$

**Lemma 3.** *Let* $\mathcal{U}$ *be an adversary and* $\mathcal{H}$ *be a hypothesis class with* $d_G^{\mathcal{U}}(\mathcal{H}) \leq d_G < \infty$. *Let* $\tilde{\mathcal{H}}$ *be the corrupted hypothesis class induced by* $\mathcal{U}$. *Let* $S = \{x_1, \dots, x_m\} \subset \mathcal{X}$. *Then, for all* $m > d_G + 1$ *we have*

$$|\tilde{\mathcal{H}}|_S| \leq \left( \frac{em}{d_G} \right)^{d_G}.$$

*Proof.* By Sauer's Lemma [26] and the fact that $\text{VC}(\mathcal{H}_f) \leq d_G^{\mathcal{U}}(\mathcal{H})$ for all $f \in \mathcal{Y}^S$. $\square$

We now prove Theorem 8. The construction idea follows the proof of the non-adversarial results in [21].

*Proof of Theorem 8:* The lower bound follows from Theorem 7 and the fact that $d_G^{\mathcal{U}}(\mathcal{H}) \geq d_N^{\mathcal{U}}(\mathcal{H})$ for all $\mathcal{H}$. To upper bound $d_G := d_G^{\mathcal{U}}(\mathcal{H})$, let $S = (x_1, \dots, x_{d_G}) \in \mathbb{R}^d$ be a set which is adversarially G-shattered by $\mathcal{H}_\Psi$, and let $f : S \to \mathcal{Y}$ be a function that witnesses the shattering. For every $(i, j) \in [d_G] \times [k]$, define $z_{i,j} = \Psi(x_i, f(x_i)) - \Psi(x_i, j)$. Denote $Z = \{z_{i,j} | (i, j) \in [d_G] \times [k]\}$. We now show that there exists a injective mapping from subsets of $S$ to $\mathcal{W}_f^{dk}|_Z$, where $\mathcal{W}^{dk}$ is the class of halfspace classifiers over $\mathbb{R}^{dk}$.

For each $T \subset S$, by the definition of adversarial G-shattering, $\exists h_T \in \mathcal{H}_\Psi$ such that

$$\forall x \in T, \forall x' \in \mathcal{U}(x), h_T(x') = f(x), \quad \text{and} \quad \forall x \in S \backslash T, \exists x' \in \mathcal{U}(x), h_T(x') \neq f(x). \quad (4)$$

Let $w_T \in \mathbb{R}^{dk}$ be the vector defines $h_T$, 4 implies

$$\forall x \in T, \forall x' \in \mathcal{U}(x), \forall j \in [k], \langle w_T, \Psi(x', f(x)) \rangle \geq \langle w_T, \Psi(x', j) \rangle,$$

and

$$\forall x \in S \backslash T, \exists x' \in \mathcal{U}(x), \exists j \in [k], \langle w_T, \Psi(x', f(x)) \rangle < \langle w_T, \Psi(x', j) \rangle. \quad (5)$$

It can be easily derived from 5 that $\forall i \in I_{S \backslash T}, \exists j \in [k], \exists z' \in \mathcal{V}(z_{i,j})$ s.t. $\langle w_T, z' \rangle < 0$, where $\mathcal{V}$ is the $p$-norm ball in $\mathbb{R}^{dk}$ with radius $r$. And from 5 we can assume w.l.o.g that $\forall i \in I_T, \forall j \in [k], \forall z' \in \mathcal{V}(z_{i,j}), \langle w_T, z' \rangle \geq 0$. This is because if there exists some $z' \in \mathcal{V}(z)$ s.t. $\langle w_T, z' \rangle < 0$, notice that $z' = \mathcal{V}(c') + z_{i,j}$ for some bounded $c'$, we can replace $x_i$ by some $x_i'$ without destroying the adversarial G-shattering (see the trick used in proof of Theorem 7), to ensure $\langle w_T, \Psi(x_i', f(x_i)) - \Psi(x_i', j) \rangle >$

$\langle w_T, c \rangle, \forall c \in \mathcal{V}(0)$. Consequently, for every $T \in S$ we obtain a label pattern $P_T : [d_G] \times [k] \to \{0, 1\}$ :

$$P_T(z_{i,j}) = \mathbb{1}(\min_{z' \in \mathcal{V}(z_{i,j})} \langle w_T, z' \rangle \geq 0).$$

It satisfies that $\forall i \in I_T, j \in [k], P_S(i, j) = 1$ and $\forall i \in I_{S \setminus T}, \exists j \in [k], P_T(i, j) = 0$, which implies $P_T \neq P_{T'}$ for $T \neq T'$. And by definition $P_T$ is connected with a unique element in $\mathcal{W}_f^{dk}|_Z$, which completes the proof of our claim.

Note that the AG-dimension of halfspace classifier class with respect to $\mathcal{U}$ coincides with the definition of adversarial VC-dimension in [14]. Thus it satisfies that $d_G^{\mathcal{U}}(\mathcal{W}^{dk}) = dk + 1$ by Theorem 2 in [14]. Consequently, we have

$$|2^S| = 2^{d_G} \leq \left| \mathcal{W}_f^{dk}|_Z \right| \leq \left| \widetilde{\mathcal{W}^{dk}}|_Z \right| \stackrel{(i)}{\leq} \left( \frac{e|Z|}{d_G^{\mathcal{U}}(\mathcal{W}^{dk})} \right)^{d_G^{\mathcal{U}}(\mathcal{W}^{dk})} \leq (kd_G)^{dk+1},$$

where we have used Lemma 3 in step (i). We conclude that $d_G \leq O(dk \ln dk)$. $\qquad\square$

## 6 AG-dimension and AN-dimension are not equivalent

In previous sections, we have shown that the finiteness of the AN-dimension is a necessary condition for robust learnability, and the finiteness of the AG-dimension is a sufficient condition for robust learnability. Furthermore, we analyze these definitions on the linear multiclass predictor class, finding that the AG-dimension (AN-dimension) matches the order of the graph dimension (Natarajan dimension) for this class. The following question therefore naturally arises: are these two quantities equivalent? In [12], it is proven that for every hypotheses class $\mathcal{H} \subset \mathcal{Y}^{\mathcal{X}}$,

$$d_N(\mathcal{H}) \leq d_G(\mathcal{H}) \leq 4.67 \log_2(k) d_N(\mathcal{H}). \tag{6}$$

That is, in standard learning setting, if $k \leq \infty$ then the definition of the Natarajan dimension and the graph dimension are indeed equivalent, hence both the Natarajan dimension and graph dimension can characterize the standard learnability of a hypothesis class. However, in the adversarial setting ($\mathcal{U} \neq \mathcal{I}$), the generalized definitions derived above cannot bound each other. To show this, we claim that finite AN-dimension is not sufficient for robust learnability.

**Theorem 9.** *There exist $\mathcal{H}, \mathcal{U}$ such that $d_N^{\mathcal{U}}(\mathcal{H}) = 0$ but $\mathcal{H}$ is not robustly PAC learnable under $\mathcal{U}$, i.e. $m_{\mathcal{H}, \mathcal{U}}^{ag}(\epsilon, \delta) = \infty$.*

*Proof.* Let $\mathcal{U}$ be the 2-norm ball: $\mathcal{U}(x) = \{z \in \mathcal{X} : \|z - x\|_2 \leq r\}$ for all $x \in \mathcal{X}$ and some $r > 0$. We construct $\mathcal{H}$ as follow. Pick a sequence $\{x_i\}_{i=1}^{\infty} \subset \mathcal{X}$ such that for any $i \neq j, \mathcal{U}(x_i) \cap \mathcal{U}(x_j) = \emptyset$. Let $f : \mathcal{X} \to \mathcal{Y}$ be a function satisfying: there exists $z_i \in \mathcal{U}(x_i) \setminus x_i$ such that $f(z_i) \neq f(x_i)$ for each $i \in \mathbb{N}$. For each sequence $b = \{b_i\}_{i=1}^{\infty} \subset \{0, 1\}$, define $h_b : \mathcal{X} \to \mathcal{Y}$ :

$$h_b(x) = \begin{cases} f(x_i), & x \in \mathcal{U}(x_i) \setminus z_i, \quad i \in \mathbb{N}, \\ f(x_i), & x = z_i \text{ and } b_i = 1, \\ f(z_i), & x = z_i \text{ and } b_i = 0, \\ 1, & \text{otherwise.} \end{cases}$$

Let $\mathcal{H} = \{h_b : b \text{ is a sequence in } \{0, 1\}\}$. Using the same approach employed in the proof of Lemma 1, for any learning rule $\mathcal{A} : \cup_{n=0}^{\infty} (\mathcal{X} \times \mathcal{Y})^n \to \mathcal{H}$, there exists a distribution $\mathcal{D}$ over $\mathcal{X} \times \mathcal{Y}$ such that there exists a function $f^* \in \mathcal{H}$ with $R_{\mathcal{U}}(f^*; \mathcal{D}) = 0$, and with probability of at least $1/7$ over the choice of $\mathcal{S} \sim \mathcal{D}^m$ we have that $R_{\mathcal{U}}(\mathcal{A}(\mathcal{S}); \mathcal{D}) \geq 1/8$, by considering the distribution family $\mathfrak{D}$ containing distributions that are uniform over $2m$ points in $\{(x_1, f(x_1)), \ldots, (x_{3m}, f(x_{3m}))\}$. We complete the proof by noting that $d_N^{\mathcal{U}}(\mathcal{H}) = 0$, since there is no single $r$-ball that is adversarially N-shattered by $\mathcal{H}$. $\qquad\square$

## 7 Conclusion

In this work, we construct the concepts of AG-dimension and AN-dimension to upper bound and lower bound the sample complexity of robust multiclass learning, respectively. We further analyze the AG/AN-dimension of linear multiclass predictors and prove that AG-dimension and AN-dimension are not equivalent in general. However, establishing a complexity measure that characterizes robust learnability still remains an open question.

## Acknowledgements

This work is supported by the National Natural Science Foundation of China under Grant 61976161.

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
