# A Appendix

We present proofs omitted from the main text here.

**Lemma 1.** *Let $m \in \mathbb{N}$. Then, there exists a hypothesis class $\mathcal{H} \subset \mathcal{Y}^{\mathcal{X}}$ such that for any learning rule $\mathcal{A} : \cup_{n=0}^{\infty}(\mathcal{X} \times \mathcal{Y})^n \to \mathcal{H}$, there exists a distribution $\mathcal{D}$ over $\mathcal{X} \times \mathcal{Y}$ such that:*
*1. There exists a function $f^* \in \mathcal{H}$ with $R_{\mathcal{U}}(f^*; \mathcal{D}) = 0$.*
*2. With probability of at least $1/7$ over the choice of $\mathcal{S} \sim \mathcal{D}^m$ we have that $R_{\mathcal{U}}(\mathcal{A}(\mathcal{S}); \mathcal{D}) \geq 1/8$.*

*Proof.* The proof follows from Lemma 3 in [13]. We construct $\mathcal{H}_0$ as follow. Pick $3m$ points $x_1, \ldots, x_{3m}$ in $\mathcal{X}$ such that for all $i, j \in [3m], \mathcal{U}(x_i) \cap \mathcal{U}(x_j) = \emptyset$. For each $b \in \{0,1\}^{3m}$, we construct a set $\mathcal{Z}_b$ : Initialize $\mathcal{Z}_b = \emptyset$, for each $i \in [3m]$, if $b_i = 1$ then pick a point $z \in \mathcal{U}(x_i)$ such that $z \notin \mathcal{Z}_{b'}$ for each $b' \neq b$, and add it to $\mathcal{Z}_b$. Let $h_b : \mathcal{X} \to \mathcal{Y}$ be a hypothesis such that $h_b(x) = 1$ if and only if $x \notin \mathcal{Z}_b$. Then, $\mathcal{H}_0 = \{h_b : b \in \{0,1\}^{3m}\}$. Consider a subset of $\mathcal{H}_0$:

$$\mathcal{H} \triangleq \{h_b \in \mathcal{H}_0 : \sum_{i=1}^{3m} b_i = m\}$$

and a family of distributions $\mathfrak{D} \triangleq \{\mathcal{D}_1, \ldots, \mathcal{D}_T\}$, where $T = \binom{3m}{2m}$ and $\mathcal{D}_i$ is uniform over only $2m$ points in $\{(x_1, 1), \ldots, (x_{3m}, 1)\} \triangleq C$ for each $i = 1, \ldots, T$. For every distribution $\mathcal{D}_i$, there exists a classifier $h^* \in \mathcal{H}$ such that $R_{\mathcal{U}}(h^*; \mathcal{D}_i) = 0$. We now prove that there exists a distribution $\mathcal{D}_r$ such that

$$\mathbb{E}_{\mathcal{S} \sim \mathcal{D}_r^m}[R_{\mathcal{U}}(\mathcal{A}(\mathcal{S}); \mathcal{D}_r)] \geq \frac{1}{4}.$$

To show this, we pick an arbitrary sequence $S \subset C$ with size $m$. Denote by $E_{\mathcal{S}}$ the event that $\mathcal{S} \subset \mathrm{supp}(\mathcal{D}_j)$, where $\mathcal{D}_j$ is a randomly picked distribution from $\mathfrak{D}$. We first lower bound the expected robust loss of the classifier that rule $\mathcal{A}$ outputs, namely $\mathcal{A}(\mathcal{S})$, given the event $E_{\mathcal{S}}$,

$$\mathbb{E}_{\mathcal{D}_i}[R_{\mathcal{U}}(\mathcal{A}(\mathcal{S}); \mathcal{D}_i)|E_{\mathcal{S}}] = \mathbb{E}_{\mathcal{D}_i}\left[\mathbb{E}_{(x,y) \sim \mathcal{D}_i}\left[\sup_{x' \in \mathcal{U}(x)} \mathbb{1}[\mathcal{A}(\mathcal{S})(x') \neq y]\right]\Bigg| E_{\mathcal{S}}\right]. \tag{7}$$

By law of total probability, we have

$$\mathbb{E}_{(x,y) \sim \mathcal{D}_i}\left[\sup_{x' \in \mathcal{U}(x)} \mathbb{1}[\mathcal{A}(\mathcal{S})(x') \neq y]\right]$$
$$\geq \mathbb{P}_{(x,y) \sim \mathcal{D}_i}[E_{(x,y) \notin \mathcal{S}}]\mathbb{E}_{(x,y) \sim \mathcal{D}_i}\left[\sup_{x' \in \mathcal{U}(x)} \mathbb{1}[\mathcal{A}(\mathcal{S})(x') \neq y]|E_{(x,y) \notin \mathcal{S}}\right]. \tag{8}$$

Since $|\mathcal{S}| = m$, and $\mathcal{D}_i$ is uniform over its support of size $2m$,

$$\mathbb{P}_{(x,y) \sim \mathcal{D}_i}[E_{(x,y) \notin \mathcal{S}}] \geq \frac{1}{2}. \tag{9}$$

Plug 8 and 9 into 7, we have

$$\mathbb{E}_{\mathcal{D}_i}[R_{\mathcal{U}}(\mathcal{A}(\mathcal{S}); \mathcal{D}_i)|E_{\mathcal{S}}] \geq \frac{1}{2}\mathbb{E}_{\mathcal{D}_i}\left[\mathbb{E}_{(x,y) \sim \mathcal{D}_i}\left[\sup_{x' \in \mathcal{U}(x)} \mathbb{1}[\mathcal{A}(\mathcal{S})(x') \neq y]|E_{(x,y) \notin \mathcal{S}}\right]\Bigg| E_{\mathcal{S}}\right].$$

Since $\mathcal{A}(\mathcal{S}) \in \mathcal{H}$, by construction of $\mathcal{H}$, there are at least $m$ points in $C$ where $\mathcal{A}(\mathcal{S})$ is not robustly correct. Hence we can unroll the expectation over $\mathcal{D}_i$ as follows

$$\mathbb{E}_{\mathcal{D}_i}\left[\mathbb{E}_{(x,y) \sim \mathcal{D}_i}\left[\sup_{x' \in \mathcal{U}(x)} \mathbb{1}[\mathcal{A}(\mathcal{S})(x') \neq y]|E_{(x,y) \notin \mathcal{S}}\right]\Bigg| E_{\mathcal{S}}\right]$$
$$\geq \frac{1}{m} \sum_{(x,y) \notin \mathcal{S}} \mathbb{E}_{\mathcal{D}_i}[\mathbb{1}_{(x,y) \in \mathrm{supp}(\mathcal{D}_i)}|E_{\mathcal{S}}] \sup_{x' \in \mathcal{U}(x)} \mathbb{1}[\mathcal{A}(\mathcal{S})(x') \neq y] \tag{10}$$
$$\overset{(i)}{\geq} \frac{1}{m} \sum_{(x,y) \notin \mathcal{S}} \frac{1}{2} \sup_{x' \in \mathcal{U}(x)} \mathbb{1}[\mathcal{A}(\mathcal{S})(x') \neq y] \overset{(ii)}{\geq} \frac{1}{2},$$

where step (i) use the fact that $\mathbb{E}_{\mathcal{D}_i}[\mathbb{1}_{(x,y)\in\text{supp}(\mathcal{D}_i)}|E_{\mathcal{S}}] = \frac{1}{2}$, since for every $(x,y) \notin \mathcal{S}$, there are exactly half of the distributions in $\{\mathcal{D} \in \mathfrak{D}|E_{\mathcal{S}}\}$ whose supports contain $(x,y)$. And in step (ii), for every point $(x,y) \notin \mathcal{S}$, we have $\sup_{x'\in\mathcal{U}(x)} \mathbb{1}[\mathcal{A}(\mathcal{S})(x') \neq y] = 1$.

Thus it follows by 10 that $\mathbb{E}_{\mathcal{D}_i}[R_{\mathcal{U}}(\mathcal{A}(\mathcal{S}); \mathcal{D}_i)|E_{\mathcal{S}}] \geq \frac{1}{4}$. By law of total expectation,

$$\mathbb{E}_{\mathcal{D}_i}\left[\mathbb{E}_{\mathcal{S}\sim\mathcal{D}_i^m}[R_{\mathcal{U}}(\mathcal{A}(\mathcal{S}); \mathcal{D}_i)]\right] = \mathbb{E}_{\mathcal{S}\sim\mathcal{D}_i}\left[\mathbb{E}_{\mathcal{D}_i}[R_{\mathcal{U}}(\mathcal{A}(\mathcal{S}); \mathcal{D}_i)|E_{\mathcal{S}}]\right] \geq \frac{1}{4}.$$

This implies that there exists $r \in [3m]$ such that $\mathbb{E}_{\mathcal{S}\sim\mathcal{D}_r^m}[R_{\mathcal{U}}(\mathcal{A}(\mathcal{S}); \mathcal{D}_r)] \geq \frac{1}{4}$. By Markov's inequality,

$$\mathbb{P}_{\mathcal{S}\sim\mathcal{D}_r^m}[R_{\mathcal{U}}(\mathcal{A}(S); \mathcal{D}_r) > 1 - 7/8] \geq \frac{\mathbb{E}_{\mathcal{S}\sim\mathcal{D}_r^m}[R_{\mathcal{U}}(\mathcal{A}(\mathcal{S}); \mathcal{D}_r)] - (1 - 7/8)}{7/8} \geq \frac{1}{7},$$

which completes the proof. $\qquad\square$

**Proposition 1.** *Let $\mathcal{H} \subset \mathcal{Y}^{\mathcal{X}}$ be a hypothesis class and let $\tilde{\mathcal{H}}$ be the corrupted set of hypotheses induced by perturbation $\mathcal{U}$. Then we have*

$$d_G(\tilde{\mathcal{H}}) = d_G^{\mathcal{U}}(\mathcal{H}).$$

*Proof.* Obviously $d_G(\tilde{\mathcal{H}}) \geq d_G^{\mathcal{U}}(\mathcal{H})$ by definition. We now prove $d_G(\tilde{\mathcal{H}}) \leq d_G^{\mathcal{U}}(\mathcal{H})$, that is, let $S = \{x_1, \ldots, x_n\} \subset \mathcal{X}$ be G-shattered by $\tilde{\mathcal{H}}$, $S$ is also adversarially G-shattered by $\mathcal{H}$. Suppose $f : \mathcal{X} \to \tilde{\mathcal{Y}}$ is the function that witnesses the adversarial G-shattering of $\tilde{\mathcal{H}}$. For each $1 \leq i \leq n$, (i) if $f(x_i) = y_i \in \mathcal{Y}$, then $\tilde{g} \in \tilde{\mathcal{H}}, \tilde{g}(x_i) = y_i$ implies that $g(x') = y_i, \forall x' \in \mathcal{U}(x_i)$ and $\tilde{g}(x_i) \neq y_i$ implies that $g(x') \neq y_i, \forall x' \in \mathcal{U}(x_i)$ or $\tilde{g}(x_i) = \perp$ . Both cases imply that $\exists x' \in \mathcal{U}(x_i), g(x') \neq f(x_i)$. (ii) if $f(x_i) = \perp$, then $\tilde{g}(x_i) = \perp$ means $\exists x' \in \mathcal{U}(x_i), g(x') \neq f(x_i)$ and $\tilde{g}(x_i) \neq \perp$ means $\tilde{g}(x_i) = y_i$ for some $y_i \in \mathcal{Y}$, which implies $g(x') = y_i, \forall x' \in \mathcal{U}(x_i)$. In this case $\tilde{\mathcal{H}}$ G-shatters $S$ coincides with the definition of $\mathcal{H}$ adversarially G-shatters $S$ by replacing $T = S\backslash T$ in Definition 4. $\qquad\square$