# OpenReview forum: "On Robust Multiclass Learnability"
_NeurIPS.cc/2022/Conference — NeurIPS 2022 Accept_

### Official Review · Reviewer_u7Pp · 2022-07-11

**Rating:** 9
**Confidence:** 5
**Soundness:** 3 good
**Presentation:** 3 good
**Contribution:** 4 excellent

**Summary:**

This paper aims to characterize the PAC learnability of robust learning problem in the multi-class setting. To this end, define the AG dimension and AN dimension, and provide both upper and lower bounds for sample complexity of robust multi-class learning. Thereafter, they give the AG dimension and AN dimension of linear multi-class predictor class. Lastly, they show that AG dimension and AN dimension are not equivalent.

**Questions:**

1. I feel the notations in the proof of Theorem 3 is a little confusing. In the expression of $h_b$,  can the first and second cases be combined, since $y_x\neq 1$? I would appreciate it if you can define $h_b$ more clearly.
2. In section 4.2, the authors discuss alternatives to generalize the AN dimension, but I am confused about the claim “we choose the stronger notion” in Line 210. Why is the chosen notion stronger?



----------------
Thanks for the reply. The response has addressed my concerns well.

**Strengths And Weaknesses:**

Strength:
1.	This paper is novel in the sense that the robust multi-class learning under PAC framework is not studied in the literature.
2.	The motivation of this paper is strong.
3.	It is well-written and easy to follow. The logic of this paper is clear and natural.
4.	The theoretical proof seems to be sound based on my judgment.

Weakness:
1.	The authors rarely discuss the insight of the theorems and the construction in proofs.
2.	It seems that the theory in this paper can hardly help us foresee whether we can train a model that is robust to adversarial perturbations.

---

> ### Author Response · Authors · 2022-08-01
> **Comments to reviewer u7Pp**
>
> Thank you for your job in reviewing our paper.
>
> > 1. I feel the notations in the proof of Theorem 3 is a little confusing. In the expression of $h_b$, can the first and second cases be combined, since $y_x\neq 1$? I would appreciate it if you can define $h_b$ more clearly
>
> No, the construction should guarantee when there exists $m'\neq m$ s.t. $x\in X_{m'}, h_b(x)$ is always equal to a same value for all $b.$
>
> > 2. In section 4.2, the authors discuss alternatives to generalize the AN dimension, but I am confused about the claim “we choose the stronger notion” in Line 210. Why is the chosen notion stronger?
>
> Sorry for the confusion caused by our presentation. The notion we choose, namely the AN-dimension defined in our paper, is stronger because the other shattering notion mentioned in the discussion can be derived by adversarially N-shattering. And the lower bound derived by the AN-dimension is tighter than that one.

---

### Official Review · Reviewer_A37A · 2022-07-11

**Rating:** 7
**Confidence:** 2
**Soundness:** 3 good
**Presentation:** 3 good
**Contribution:** 3 good

**Summary:**

This work analyzes the robust learning problem in the multiclass setting and then derives the adversarial graph dimension (AG-dimension) and the adversarial Natarajan dimension (AN6 dimension) from the graph dimension and the Natarajan dimension.


**Questions:**

"We then analyze the AG-dimension and the AN-dimension of linear multiclass predictor class, showing that the AG(AN)-dimension and the graph (Natarajan) dimension are of the same order." can these results hold for the non-linear models?

Since the AG-dimension and AN-dimension are not equivalent, which one is better?

**Ethics Review Area:**

["I don’t know"]

**Limitations:**


Please see the comment above.


**Strengths And Weaknesses:**

This paper shows that the graph dimension and the Natarajan dimension are no longer applicable in robust learning problem and then generalize these notions to the robust learning setting, denoted as the adversarial graph dimension (AG-dimension) and the adversarial Natarajan dimension (AN dimension).

If there provide some experimental results to verify the theoretical analysis will be more convincing.

---

> ### Author Response · Authors · 2022-08-01
> **Comments to reviewer A37A**
>
> Thank you for your job in reviewing our paper.
>
> > "We then analyze the AG-dimension and the AN-dimension of linear multiclass predictor class, showing that the AG(AN)-dimension and the graph (Natarajan) dimension are of the same order." can these results hold for the non-linear models?
>
> The answer is no. In the proof of Theorem 9, we have constructed a hypothesis class $\mathcal{H}$, whose members are not linear, with $d_N^{\mathcal{U}}=0$ but $d_G^\mathcal{U}=\infty.$ But we think it is of theoretical interest to study if these results hold for some specific models, e.g. NN, for follow-up work.
>
> > Since the AG-dimension and AN-dimension are not equivalent, which one is better?
>
> AG-dimension and AN-dimension are two notions to characterize the upper and lower bounds of the sample complexity for robust multiclass learning, respectively. We think they are both important to perfect the theory in this area.

---

### Official Review · Reviewer_PJsj · 2022-07-20

**Rating:** 8
**Confidence:** 3
**Soundness:** 4 excellent
**Presentation:** 4 excellent
**Contribution:** 3 good

**Summary:**


This work considers the adversarially robust PAC-learnability in the context of muulticlass learning. The main contributions are the following.

- First the authors show that the usual G-dimension ("G" for "graph") and N-dimension ("N" for "Natarajan) notions of complexity which are sufficient to gaurantee ordinary multiclass PAC-learnability, are in general, neither sufficient nor necessary nor adversarial PAC-learnability. Refer to Theorems 3 and 5.

- Then, the authors extend the G-dimension / N-dimension complexity measures to the adversarial setting, namely: adversarial G-dimension, or AG-dimension for short, and adversarial N-dimension, or AN-dimension for short. The authors show that finite AG-dimension (resp. finite AN-dimension) is sufficient (resp. necessary) for adversarial multiclass PAC-learnability. See Theorems 4 and 6.


- For bounded $\ell_p$-norm attacks, the AG-dimension and AN-dimension of linear multiclass predictors is computed to be rough $\tilde \Theta (d\cdot k)$, where $d$ is the input-dimension and $k$ is the number of classes. See Theorems 7 and 8.

- Finally, the authors prove (Theorem 9) that AG-dimension and AN-dimension are not equivalent because unlike AG-dimension, finite AN-dimension is not sufficient for adversarial PAC-learnability. This is in stack contrast with the ordinary / non-adversarial scenario finiteness of G-dimension and N-dimension each fully characterize PAC-learnability, and so, these dimensions are equivalent to each  other.

**Questions:**

For the binary-case where $k=2$, do the results presented in this paper fully recover the existing literature ?

**Limitations:**

This is a purely theoretical work about understanding the algorithmic gap between ordinary and adversarial learnability. There are no social / societal issues here.

**Strengths And Weaknesses:**

Strengths
---
- First work to fully characterize multiclass PAC-learnability in adversarial setting.
- Paper is extremely well-written and contributions are clearly developed. Also ordinary / non-adversarial multiclass PAC-learnability theory is conveniently reviewed before presenting main contributions. This helps the reader crispy see the gaps between ordinary and adversarial theory (developed here).

Weaknesses
---
I could find no real weaknesses, just a few typos, etc.:

- Line 73: "$\hat D_n$ is equal to $(x_i,y_i)$ w.p $1/n$" doesn't seem to make sense. Indeed, $\hat D_n$ is a (random) distribution on $X \times Y$, while $(x_i,y_i)$ is a point of $X \times Y$.

---

> ### Author Response · Authors · 2022-08-01
> **Comments to reviewer PJsj**
>
> Thanks for your job in reviewing our paper. We have fixed the typos in revision (see Line 73).
>
> > For the binary-case where $k=2$, do the results presented in this paper fully recover the existing literature ?
>
> To our knowledge, under the framework of PAC learnability and proper learning, yes. Specifically, [7] defines the adversarial VC-dimension, which is equivalent to the AG-dimension when $k=2,$ and their main results, namely Theorems 1&2, matches our results, i.e. Thms 4&7, respectively. [6] finds that finite VC dimension cannot ensure robust learning in binary setting, we generalizes this result in Theorem 3.

---

### Official Review · Reviewer_mVKM · 2022-07-21

**Rating:** 5
**Confidence:** 4
**Soundness:** 3 good
**Presentation:** 3 good
**Contribution:** 2 fair

**Summary:**

The paper defines multiclass robust learnability and makes progress towards obtaining necessary and sufficient conditions on hypothesis classes for it. Adversarial versions of graph and Natarajan dimensions are defined to this end and used to give sufficient and necessary conditions respectively. It is also shown that the known non-adversarial graph and Natarajan dimensions do not give the respective learnability conditions. Near tight estimates of the proposed dimensions are given for linear multiclass predictors under bounded Lp perturbations, the bounds happen to match known bounds for the non-adversarial variants. Finally, it is shown that (in contrast to their non-adversarial versions) the proposed adversarial dimensions are not equivalent (for finite number of classes).


**Questions:**

Questions for the authors:

There is a subtle difference in the quantifiers over U(x) in the definitions of adversarial graph and Natarajan dimensions (existential in AG but forall in AN). Can the authors give some intuition for this difference?

Theorem 6: What is the definition of $m_{A,U}^{re}$ where A is a learning algorithm instead of a hypothesis class? In the proof of Theorem 6, how Theorem 2 is used?

Is it correct to conclude from Section 5 that sample complexity for robust and standard learning of linear predictors is about the same?

Line 280: should it be $m^{re}_{H,U} = \infty$ since the proof has $R_U(f^*,D)=0$?

Other comments/typos:
Proof of Theorem 7 repeatedly uses ideas from Theorem 8, consider switching them around for readability?


Line 71: worth commenting that only proper learning is considered, improper learning rules can output any function in Y^X not just H.

Line 133: statement is somewhat unreadable. Do you mean "if " x_i\ne x_j "then" U(x_i)\capU(x_j)=\emptyset?

Theorem 8: It seems more informative to state the theorem as 'robust natarajan dimension coincides with its non-adversarial counterpart for linear classifiers under bounded lp perturbation', and the current statement as a corollary of this result and previously known bounds.

P(x) should be U(x) in equation (1)
Line 224: S' \in (Y \times {0,1})^m
Lines 284,285 {0,1}^*
Line 290: "adversarially" N-shattered


**Limitations:**

It could be worth stating explicity that as a consequence of Theorem 9 this work does not fully characterize robust multiclass learnability and determining necessary and sufficient conditions is an open problem.

**Strengths And Weaknesses:**

Robust learnability in the multiclass setting captures an interesting and relevant learning setting. The paper provides a necessary and a sufficient condition for general hypothesis classes and also provides instantiations for linear classifiers.

An important contribution is defining the adversarial graph and natarajan dimensions, which are used to show upper and lower bounds on sample complexity of robust multiclass learning. The results seem sound.

Theorem 9 is interesting, it shows an example where the robust graph dimension is infinite but the robust Natarajan dimension is finite/zero. This implies that multiclass robust learnability is not completely characterized in this work.

My major concern is about the presentation of the results - in particular inadequate referencing to known proofs/techniques - which also makes it harder to assess the originality of the contribution. This is particularly important for this work since a lot of the presented results have known two-class and/or non-adversarial counterparts.

Some concrete examples about the reference to prior work in some of the results:
Lemma 1: The lemma looks identical to Lemma 3 of [6], even the proof is the same. You should simply cite the original lemma in this case.
Typos: should be H\subseteq Y^X instead of H\in Y^X. The expression A(S) in (2.) should have calligraphic A and S.
Theorem 3 looks like an extension of Theorem 1 of [6]. It is worth pointing out the main differences in the proof technique (i.e. in construction of h_b and bounding its graph dimension) relative to Theorem 1, [6] since the overall idea looks very similar.
Theorem 5 extends the construction in section 5 (para 1) of [6] which shows finite VC dimension is not necessary for (two-class) robust learnability.

Also, in my opinion, the novel proof/technical insights relative to prior work are not properly indicated/summarized. For example in section 5, it would be nice to summarize the novel ideas needed to extend/combine the non-adversarial results of [14] and k=2 results of [7].

---

> ### Author Response · Authors · 2022-08-01
> **Comments to reviewer mVKM**
>
> Thank you for your job in reviewing our paper. We are very sorry for the inconvenience caused by our presentations. To this end, following your comments, we have supplemented some citations on our techniques, and discussed some limitations in revision (see text highlighted in red). Typos are also fixed.
>
> >There is a subtle difference in the quantifiers over U(x) in the definitions of adversarial graph and Natarajan dimensions (existential in AG but forall in AN). Can the authors give some intuition for this difference?
>
> In the definition of AG dimension, we only need one label pattern $f$, and intuitively, AG shattering is equivalent to shattering in terms of the (robust) loss class induced by this label pattern of $\mathcal{H}$, i.e.$L_f=\{(x,y)\mapsto\sup_{x'\in \mathcal{U}(x)} l(h(x'),f(x)):h\in\mathcal{H}\}$ (see Section 3.2 in [7] for details). The existence of $x'\in\mathcal{U}(x)$ and $h\in\mathcal{H}$ s.t. $h(x')\neq f(x)$ provides the $``1"$ label. While in the definition of AN dimension, we need two different label patterns $f_1\neq f_2$. And AN-shattering is equivalent to shattering the whole perturbation sets labeled by $f_1$ and $f_2$.
>
> >Theorem 6: What is the definition of $m_{A,U}^{re}$ where A is a learning algorithm instead of a hypothesis class? In the proof of Theorem 6, how Theorem 2 is used?
>
> Thank you for checking our proofs so carefully, we are very sorry that we make some notation mistakes here. We mistakenly wrote $[d]$ as $\mathcal{Y}$ in the construction of $H_d.$ We have revised this proof and explained in Line 220 how we use, in fact, Theorem 1 in revision. Besides, the notation $m_{\mathcal{A},\mathcal{U}}^{re}$ emphasizes that the sample complexity is with respect to the algorithm $\mathcal{A},$ which characterizes how many samples $\mathcal{A}$ should witness to ensure adversarial risk $\leq\epsilon.$ You can also think of it as we omit the subscript $\mathcal{H}$ here. The common notation for sample complexity, e.g. $m_{\mathcal{H},\mathcal{U}}^{re}$, can thus be defined by $\min_\mathcal{A}m_{\mathcal{A},\mathcal{U}}^{re}$. It does not specify an algorithm, which means it needs only the existence of some algorithms that reach the risk bound.
>
> >Is it correct to conclude from Section 5 that sample complexity for robust and standard learning of linear predictors is about the same?
>
> According to our results, under the PAC framework, the sample complexity for robust and standard learning of leaner predictors is about the same. However, in most cases there will be a gap between their approximation errors, so we cannot expect the same generalization errors with the same size of samples.
>
> >Line 280: should it be $m_{H,U}^{re}=\infty$ since the proof has $R_U(f^*,D)=0$?
>
> It is okay to use $m_{H,U}^{re}$ in Theorem 9. But we think the agnostic setting is more generalized, it involves the realizable setting. We may still use $m^{ag}_{\mathcal{H},\mathcal{U}}$ here.
>
> > Proof of Theorem 7 repeatedly uses ideas from Theorem 8, consider switching them around for readability?
> >Theorem 8: It seems more informative to state the theorem as 'robust natarajan dimension coincides with its non-adversarial counterpart for linear classifiers under bounded lp perturbation', and the current statement as a corollary of this result and previously known bounds.
>
> We totally agree, and have re-organized Section 5 in revision. Specifically, we have switched Thm 7 and Thm 8 around, and made the current statement as a corollary in Lines 238-247. Thanks for these suggestions!
>
> >Line 71: worth commenting that only proper learning is considered, improper learning rules can output any function in Y^X not just H.
>
> We have pointed out that we focus on proper learning in Line 69.
>
> >Line 133: statement is somewhat unreadable. Do you mean "if " x_i\ne x_j "then" U(x_i)\capU(x_j)=\emptyset?
>
> That is what we mean. It’s much easier to read in your way, and we have taken this expression in revision (see Line 135); sorry for the confusion.
>
> >P(x) should be U(x) in equation (1) Line 224: S' \in (Y \times {0,1})^m Lines 284,285 {0,1}^* Line 290: "adversarially" N-shattered
>
> Thanks for the checking, these typos have been fixed in revision (see Eq(1), Line 224, Line 294, respectively).

---

> > ### Comment · Reviewer_mVKM · 2022-08-06
> > **Post rebuttal comments**
> >
> > Thank you for answering my queries and taking into account the feedback. The presentation of the results is clearer to me in the author's revision, in particular there is more clear referencing to prior work showing where known results have been generalized and extended. I am increasing my score as a result.
> >
> > I believe there is still scope for improving the readability by summarizing at a high level the novel proof/technical insights and challenges relative to prior work. As the authors confirm in their revision, most of the results are obtained by extending corresponding binary class or non-adversarial results to the multi-class and robust setting respectively.

---

> > > ### Author Response · Authors · 2022-08-07
> > > **Re: Post rebuttal comments**
> > >
> > > We appreciate the feedback of reviewer mVKM. To further improve the presentation, we have added a Technique Overview subsection 1.1 in revision (see Lines 41-77 highlighted in red). In the overview, we summarize at a high level the proof/technical insights and challenges relative to prior work:
> > >
> > > - Our Theorems 3 and 5, stating that the finiteness of graph dimension is not sufficient and the finiteness of Natarajan dimension is not necessary for robust learning, respectively, are demonstrated by constructing counter examples. The arguments follow closely the proofs of analogous results but generalize the constructions and analyzes to match the definition of graph/Natarajan dimension in the multiclass setting.
> > > - Our main results, namely Theorems 4 and 6, are high probability bounds of sample complexity that are sufficient/necessary for robust PAC learning, respectively. Compared to standard PAC learning, robust PAC learning is mainly different in the definition of loss function, which considers the maximum of 0-1 loss over a perturbation set. Thus to prove these bounds, we need to design a way that takes the disturbance set into account to control the adversarial risk in a high probability. [7] presents the notion of corrupted hypothesis class and use it to derive an upper bound for binary robust learning by considering the VC-dimension of the corrupted hypothesis class. This approach shed some light on our work. However, generalizing to multiclass setting in this way faces some difficulties, since there are quite a few definitions that can be generalized to multiclass setting, e.g. corrupted hypothesis class, shattering coefficient and VC-dimension in the context of adversarial setting. To make it more natural, we define the corrupted hypothesis class in multiclass setting, which does not involve fixed label patterns, and then study the graph/Natarajan dimension of a corrupted hypothesis class. For clarity, we analytically abstract these definitions as the AG/AN-dimension of hypothesis class, making it get rid of the notion of corrupted hypothesis class. There are some beneficial efforts by defining in this way: one is to intuitively correspond the graph/Natarajan dimension to the AG/AN-dimension, as well as the bounds derived by these notions. Another benefit is to make it possible to transfer common multiclass proof techniques to adversarial setting, namely constructing auxiliary binary classification problems and using them to control the risk we focus on.
> > > - Theorems 7 and 8 give bounds of AG/AN-dimension for linear multiclass predictors $\mathcal{H}_\Psi$. To show these results, we need to prove two non-trivial bounds, namely the upper bound of AG-dimension and the lower bound of AN-dimension. For the upper bound, we gain our construction idea from [8], which calculates the graph dimension of multiclass SVM. Yet their proof make use of Sauer's Lemma [9], which is only applicable for standard setting. To fill this gap, we present Lemma 2 - the generalization of Sauer's Lemma in adversarial setting - showing that the growth function in the context of adversarial and multiclass setting, which is defined very finely, also increases polynomially as the number of samples $m>d_G^{\mathcal{U}}+1$, where $d_G^{\mathcal{U}}$ is the AG-dimension of some hypothesis class. As for the lower bound, we claim that for linear classifiers, its AN-dimension equals to its Natarajan dimension. We demonstrate this by geometric analysis, which makes use of the linearity of predictors in $\mathcal{H}_\Psi$.
> > > - We prove Theorem 9 by constructing a novel counter example to show the finiteness of AN-dimension is not sufficient for robust multiclass learning, which reveals that the AG-dimension and AN-dimension are not equivalent.

---

> > > ### Author Response · Authors · 2022-08-08
> > > **Discussion**
> > >
> > > Dear reviewer mVKM, thanks for your comments. Following your suggestions, we have tried our best to improve the presentation in revision. Would you please take a look about our new Subsection 1.1 in revision and provide some suggestions, if possible？Many thanks for your time.

---

> ### Author Response · Authors · 2022-08-06
> **Discussion**
>
> Dear reviewer mVKM, we will really appreciate it if the reviewer can go over our detailed response and revisions. Please feel free to ask us any questions you may still have and we will be more than happy to answer them.                                                                                                    Thank you again for reviewing our paper and we look forward to discussing with you.

---

### Meta-Review · Area_Chair_r5XA · 2022-08-24

**Recommendation:** Accept
**Confidence:** Certain

**Metareview:**

Exceptional contribution to multi-class theory

**Award:**

Yes

---

### Decision · Program_Chairs · 2022-09-14

Accept